

# Estimating the timescale-dependent uncertainty of paleoclimate records—a spectral approach. Part II: Application and interpretation.

Andrew M. Dolman[1], Torben Kunz[1], Jeroen Groeneveld[1], and Thomas Laepple[1,2]

[1]Alfred-Wegener-Institut Helmholtz-Zentrum für Polar-und Meeresforschung, Research Unit Potsdam, Telegrafenberg A45, 14473 Potsdam, Germany
[2]University of Bremen, MARUM – Center for Marine Environmental Sciences and Faculty of Geosciences, 28334 Bremen, Germany

**Correspondence:** Andrew M. Dolman (andrew.dolman@awi.de)

**Abstract.**

Proxy climate records are an invaluable source of information about the earth's climate prior to the instrumental record. The temporal- and spatial-coverage of records continues to increase, however, these records of past climate are associated with significant uncertainties due to non-climate processes that influence the recorded and measured proxy values. Generally, these

5 uncertainties are timescale-dependent and correlated in time. Accounting for structure in the errors is essential to providing realistic error estimates for smoothed or stacked records, detection of anomalies and identifying trends, but this structure is seldom accounted for. In the first of these companion articles we outlined a theoretical framework for handling proxy uncertainties by deriving the power spectrum of proxy error components from which it is possible to obtain timescale-dependent error estimates. Here in part II, we demonstrate the practical application of this theoretical framework using the example of

10 marine sediment cores. We consider how to obtain estimates for the required parameters and give examples of the application of this approach for typical marine sediment proxy records. Our new approach of estimating and providing timescale-dependent proxy errors overcomes the limitations of simplistic single value error estimates. We aim to provide the conceptual basis for a more quantitative use of paleo-records for applications such as model-data comparison, regional and global synthesis of past climate states and data assimilation.

## 1 Introduction

Proxies of climate variables, such as geochemical indicators of temperature in marine sediments or ice-cores, are a valuable source of information about the earth's climate prior to the instrumental record. However, these records are an imperfect representation of past climate as they are also influenced by non-climatic factors in addition to the climate signal. Errors

in a proxy record mean that the past climate inferred from these proxy records is uncertain; understanding these associated





uncertainties is important for all quantitative uses of climate proxies, such as data assimilation (Goosse et al., 2006), model-data comparisons (Lohmann et al., 2013), hypotheses testing (Hargreaves et al., 2011) or the optimal combination and synthesis of climate records (Marcott et al., 2013; Shakun et al., 2012). Finally, knowing the error as a function of environmental or proxy specific parameters also allows for optimisation of the sampling and measurement strategy in order to obtain the information

required to test specific hypotheses.

Errors in a proxy record, defined here as differences between the climate inferred from the proxy record and the true climate, are introduced at multiple stages between the true climate signal and final inferred past climate time-series (see for example, Evans et al., 2013; Dee et al., 2015; Dolman and Laepple, 2018). Importantly, the resulting errors are not all independent in time, rather they are often correlated and timescale-dependent (Fig. 1). Currently the temporal covariance structure of proxy

uncertainties is largely ignored in the literature. In many cases, a single number, perhaps derived from a calibration dataset, is reported as the uncertainty for a given proxy. However, its utility is very limited without additional information about the structure of the error. For example, consider an error of 1.5°C. If the error were due to an uncertainty in the temperature to proxy relationship, e.g. the error of the intercept of a linear calibration equation, the uncertainty of a time-slice containing multiple observations would still be 1.5°C, as the error does not reduce by averaging more samples calibrated using the same

equation (Fig. 1c), while the error from calibration on a difference between two time-slices would be zero. On the other hand if this error were independent in time and thus between samples, e.g. if it were related to the error of a measurement device, the uncertainty of a time-slice based on nine samples would be just $1.5°C/\sqrt{9} = 0.5°C$, while the error on the difference between two time-slices would be approximately 0.7°C ($\sqrt{2 \cdot 0.5^2}$). Indeed, a number of recent studies assume independence in time (and space) and thus arrive at unrealistic uncertainty estimates (e.g., Fedorov et al., 2013; Shakun et al., 2012; Marcott et al.,

20   2013).

More difficult than either fully independent errors (Fig. 1a), or constant errors or "biases" (Fig. 1c), are correlated errors that manifest as slowly varying biases (Fig. 1b), for which we need to quantify both the magnitude and autocorrelation structure. The idea we introduce in part I (Kunz et al., 2020) is to work in the spectral domain, as this allows an explicit representation of the timescale dependence of uncertainty. The power-spectrum of a proxy error contains all the information required to derive

timescale dependent uncertainties, and working in the frequency domain further simplifies the estimation of the different error components. A number of additional useful quantities such as the uncertainty in a time-slice mean, the uncertainty in the difference between two time-slices and the expected timescale-dependent correlation between replicates of proxy records and between proxy records and the true climate, can easily be derived from the error spectrum.

In part I we discuss the theoretical basis for and give a full mathematical derivation of the Proxy Spectral Error Model

(PSEM). Here in part II we aim to facilitate the use of PSEM in paleo-climate applications. Thus, we 1) sketch the concepts behind the different error components in a more applied way, 2) provide heuristic approaches to parametrise the climate spectra and other parameters of the error model, 3) provide examples using virtual and actual sediment cores, and 4) provide an R-package implementing the spectral error estimation method.





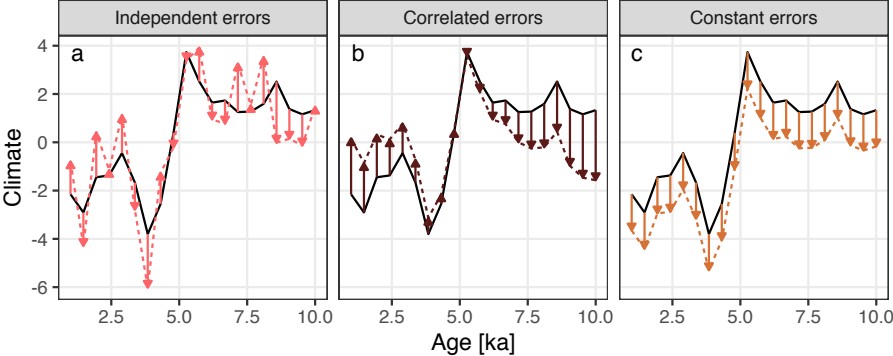

**Figure 1.** Illustration of different timescale dependence of proxy errors. The independent and correlated errors both have standard deviations of 1.5°C, while the constant error is 1.5°C with zero standard deviation.

## 2 Error spectra as a framework for timescale-dependent proxy uncertainty

In this section we illustrate the error spectrum framework for the specific example of temperature related proxies in the shells of planktic foraminifera recovered from marine sediment cores. The major processes contributing uncertainty to these proxy records have been explored using physically motivated proxy forward models that simulate pseudo-proxies from an assumed

true input climate (Laepple and Huybers, 2013; Dolman and Laepple, 2018). These processes include seasonality in the creation of proxy signal carriers (e.g. foraminiferal tests), aliasing due to under-sampling of the seasonal climate cycle, mixing and smoothing of the signal due to bioturbation, and independent measurement and processing error. One approach to estimating the uncertainty for a given metric and proxy record is to use such a forward model to simulate ensembles of pseudo-proxy records, calculate the metric for each and then examine their statistical properties. In the approach we propose here, we do not

simulate pseudo-proxies for a specific climate time-series, rather we make some simplifying assumptions about the properties of the power spectrum of the climate and then estimate the uncertainty directly from expressions for the power spectra of the errors associated with proxy creation. The advantage of this analytical method over simulation is that it allows very rapid assessment of proxy error, the relative contribution of different error sources, the expected correlation between replicated proxy records and between proxy records and the true climate, and all of these at multiple timescales. This makes it possible to scan

potential coring locations, sampling and measurement strategies to optimise future data acquisition as well as help to interpret existing records. Finally, PSEM also provides a basis for estimating the spectrum of climate variability from error corrupted proxy records at a future stage.

### 2.1 A simple model for the power spectrum of the climate

The uncertainty from some proxy error sources depends on the strength of the variations in the climate. For example, the

error from smoothing a time-series is zero if the time-series is constant and becomes larger the more the time-series varies. We therefore need a model for the variability of the climate and we describe this using the power spectrum of the climate

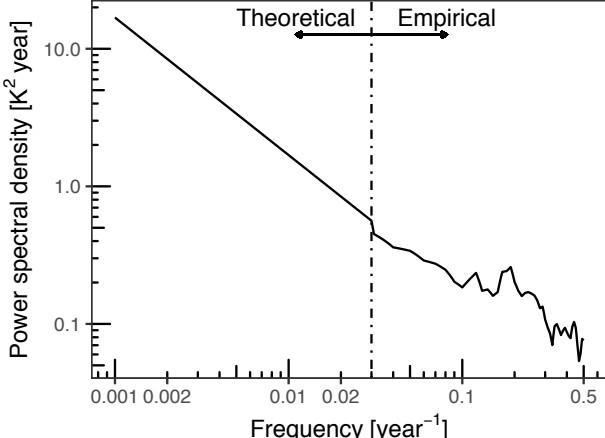

**Figure 2.** An example spliced empirical and power-law power spectrum of ocean temperature (0-120 m water depth) for 20° N latitude. A small discontinuity at $\nu = 0.03$ is visible, as to increase the robustness of the intercept estimate, the splicing is implemented by matching the integrated spectra between $\nu = 0.03$ and $\nu = 0.1$.

$S(\nu)$ (see sections 2.1 and 3.1 of part I). While the developed approach allows for any choice of a climate spectrum, e.g. from complex climate models or estimated from observations, we here outline an heuristic method suitable for marine sea-surface-temperature (SST) and surface $\delta^{18}$O calcite records. This method is implemented in the PSEM R package and requires only the core position and habitat depth range as input parameters.

A detailed, site-specific power spectrum of the climate can only easily be estimated empirically for timescales up to the length of the instrumental record. At longer time scales the climate spectrum can be approximated by a power-law type scaling, $S(\nu) = \alpha\nu^{-\beta}$, where the exponent $\beta$ characterises the scaling behaviour and is thought to lie between 0 and 2 (Lovejoy, 2015; Schmitt et al., 1995) and $\alpha$ scales the amplitude of climate variation. Here we take the pragmatic approach of splicing zonally averaged empirical climate spectra for frequencies above 1/33 years with theoretical power law spectra at lower frequencies

(Laepple and Huybers, 2014b) (Fig. 2). As the empirical spectra were estimated from annual resolution ocean temperature records, we set power to zero at frequencies above 1/2 years.

    For the power-law section of the climate spectra $\alpha$ was chosen so that the low frequency power law spectra are continuous with the empirically estimated high frequency regions of the spectra. We typically assume a value of 1 for $\beta$ as this was found to be a good description of Holocene SST variability (Laepple and Huybers, 2014a) but this parameter can be freely

specified. To allow these spectra to also be used for $\delta^{18}$O records we re-calibrate them to $\delta^{18}$O calcite units using a standard calibration (Bemis et al., 1998), which in terms of variance is effectively just a division by a factor of $4.8^2$, assuming that the $\delta^{18}O_{calcite}$ is generally dominated by temperature variations at these timescales. A function to generate these spliced empirical and power-law spectra is supplied as part of the PSEM R package.

    To this stochastic climate we add a deterministic seasonal cycle modeled as the power spectrum of a sine wave with frequency

1/1 year (see section 3.2 in part I). For a given location, we parametrise the amplitude of the seasonal cycle using a gridded





data set of assimilated physically consistent $\delta^{18}O_{sw}$ and temperature (Breitkreuz et al., 2018). $\delta^{18}O_{calcite}$ was calculated on the PDB scale from $\delta^{18}O_{sw}$ and temperature using the equations of Shackleton (1974).

Additionally, the amplitude of earth's seasonal temperature cycle varies over a precessionary orbital cycle with approximate frequency 1/23 kyr. The magnitude of this amplitude modulation depends primarily on latitude and we assume this is equal
to the amplitude modulation of incoming solar radiation at a given latitude (Berger and Loutre, 1991). This introduces a low frequency (1/23 kyr) deterministic signal to the climate.

## 2.2   Proxy error processes as filters.

During the creation of a climate proxy record, some of the processes that introduce errors, defined here as differences between the proxy record and the true climate, can be thought of as acting like filters on the true climate signal. Here we illustrate the
concept of PSEM by considering the smoothing effects of bioturbation and the width of sediment slices from which signal carriers, e.g. foraminiferal shells, are extracted.

Bioturbation at the water-sediment interface mixes the upper few centimeters of sediment thereby mixing together signal carriers of different ages. This acts like a smoothing filter on the climate signal, reducing the amplitude of climate variations in a frequency dependent way. The magnitude of the reduction at each frequency depends on the filter characteristics; using the
simple physical bioturbation model of Berger and Heath (1968), the filter width, $\tau_b$, is simply the bioturbation depth divided by the sedimentation rate. In the frequency domain, this is equivalent to multiplying the power spectrum of the climate with the transfer function of the filter and results in a power spectrum of the error as described in part I, sections 2.2 and 3.1.

Similarly, when a sediment core is sampled the proxy signal carriers are picked from a series of slices of sediment, each with a finite width of typically 1-2 cm. This again acts as a filter, this time a running mean filter with width, $\tau_s$, equal to the ratio
of the slice thickness and sedimentation rate. As for the bioturbation filter, in the frequency domain the effect on the original signal is obtained by multiplying the power spectrum with the transfer function of the filter (see sections 2.3 and 3.1 in part I).

### 2.2.1   Error relative to the reference climate.

So far we refer to a proxy error as a difference between the measured proxy value and the "true" climate signal. The bioturbation and slice thickness filters smooth the climate signal so that the proxy differs from the true climate. However, in practice, values
of proxy variables from marine sediments are rarely interpreted as representing the instantaneous climate state, it is understood implicitly that some smoothing has taken place. The error therefore depends on the interpreted timescale of the proxy record and so, in the spectral error model, we define error relative to the true climate at a specific timescale provided by the user (see section 2.4 in part I). The power-spectrum of this error is shown as the brown dashed line in Fig. 3a.

## 2.3   Redistribution of climate power due to under-sampling

The proxy quantity (e.g. $\delta^{18}O$, Mg/Ca etc.) is often measured on a finite number, $N$, of discrete signal carriers such as foraminiferal tests, each of which calcifies and records or samples a short snapshot of the climate, typically 2-4 weeks for

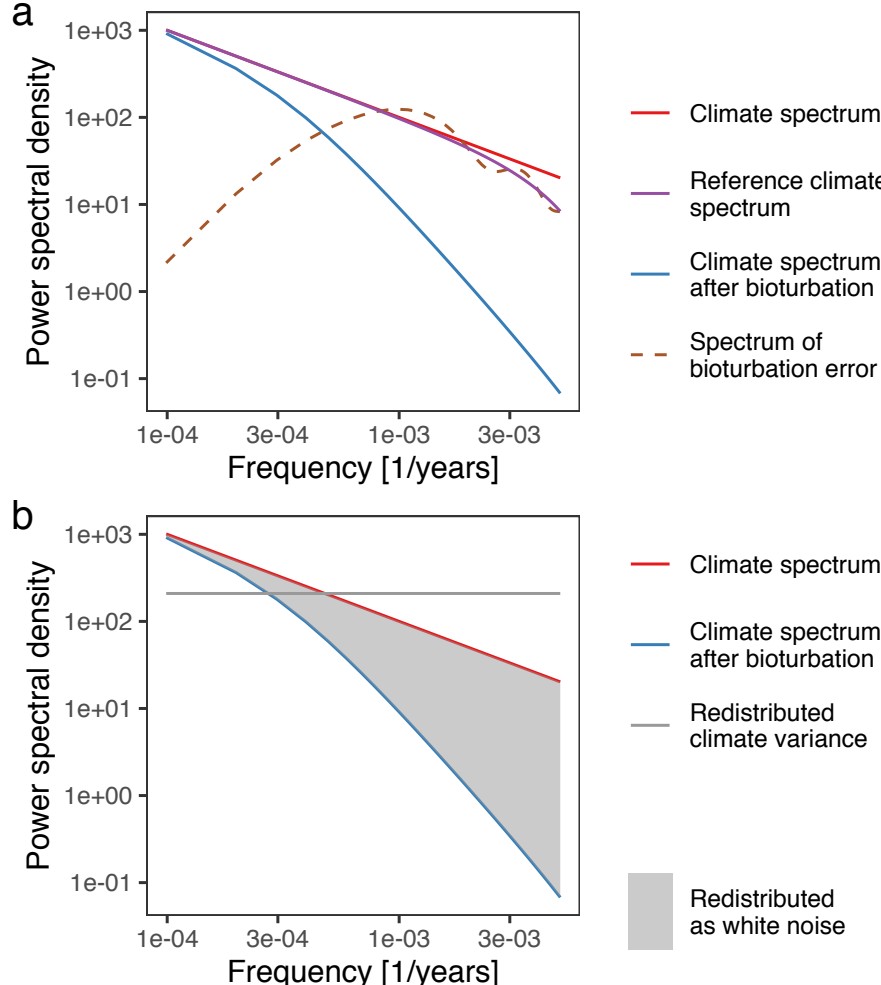

**Figure 3.** A conceptual representation of PSEM. a) The true climate signal is filtered (smoothed) by processes such as bioturbation. This modifies the power spectrum of the climate (red) in a frequency dependent way, producing the power spectrum of the climate signal after bioturbation (blue). Proxy records are assumed to represent the climate at a particular timescale, (e.g. centennial, millennial), the reference climate spectrum (purple) is the power spectrum of the true climate smoothed to this timescale. The error that bioturbation and other smoothing produces (dashed brown) is a function of the reference and bioturbated climate spectra. b) The power of the true climate signal that is removed by bioturbation and other smoothing processes (grey shaded region) can be redistributed as a white noise error component (grey horizontal line) if the time period over which individual signal carriers are created, (e.g. foraminiferal tests calcify), is short relative to the timescale of the smoothing processes. This white noise term is reduced if multiple signal carriers are measured together, or if their values are averaged together later.

pelagic foraminifera (Bijma et al., 1990; Spero, 1998). The bioturbation and slice thickness filters can be thought of as probability density functions (PDFs) that describe the portion of time from which the climate signal is sampled by these $N$ signal





carriers. As this is a finite sample, the resulting proxy value is an estimate of the mean value and contains a stochastic noise component in addition to the deterministic error caused by the smoothing. The variance of this stochastic error term is equal to the integral of the difference between the climate spectrum and the spectrum of the smoothed climate signal, divided by the number of individual signal carriers in a sample, shown as the grey line and shaded area in Fig. 3b (see sections 2.3 and 3.1 in part I). As $N$ tends to infinity, the error due to undersampling tends toward zero. This may be the case for proxies such as Uk'37, where measurements consists of many thousands of organic molecules.

## 2.4 The seasonal cycle

The often large cyclical variation in climate variables associated with the seasonal cycle can also add noise to a proxy record if the seasonal cycle is not adequately sampled by individual signal carriers, each of which records a short portion of the cycle (Laepple and Huybers, 2013; Schiffelbein and Hills, 1984). Additionally, a bias can be introduced in the record if the signal carriers are produced in greater numbers during a particular part of the year (Jonkers and Kučera, 2015; Leduc et al., 2010). In the case of orbital modulation of the amplitude of the seasonal cycle, this bias becomes a slowly varying error, and the variance of the noise process also varies over the course of an orbital cycle (see section 3.2 of part I).

In the spectral error model, the seasonal cycle is represented by the discrete power spectrum of a deterministic sine wave (Eq.1 in part I). The interaction between this signal and seasonality in the production of signal recorders determines the magnitude of two errors - a white noise error generated by undersampling of the seasonal cycle and a bias due to only sampling a portion of the seasonal cycle.

We represent seasonality in the production of signal carriers here by saying that production occurs continuously over a set fraction of the year, $\tau_p$ (see section 2.2 in part I). This can also be viewed as a kind of filter, this time on the discrete spectrum of the deterministic sine wave seasonal cycle. The transfer function of this filter can be constructed in an analogous way to those for bioturbation and slice thickness, and the difference between the filtered and unfiltered spectra gives the error due to sampling only part of the seasonal cycle (see section 3.2 in part I). Again, the finite time-period each carrier records means that the portion of the seasonal cycle during which signal carriers are created is sampled by the individuals and this generates additional redistributed white noise in the proxy signal.

In the absence of orbital modulation, four parameters determine the errors generated by filtering and sampling the seasonal cycle:

$\sigma_c^2$, the variance of the full seasonal cycle; $\tau_p$, the proportion of the seasonal cycle during which signal carriers are produced; $\langle\phi_c\rangle$, the expected midpoint (phase) of the signal carrier production; and $\Delta\phi_c$, which represents uncertainty in the phase of the carrier production (section 2.5 in part I).

If the signal carriers are produced all year round, $\tau_p = 1$, there is no bias and the white noise component has a variance equal to the variance of the full seasonal cycle divided by the number of signal carriers per sample, $N$.

If the signal carriers are produced for only part of the year, $\tau_p < 1$, but with completely unknown timing (we don't know which months, $\Delta\phi_c = 2\pi$), then the expected variance of this white noise is equal to the difference between the variance of the full seasonal cycle and the variance of the seasonal cycle filtered with a running mean filter of width, $\tau_p$. In the spectral





domain, this is analogous to the grey shaded area in Fig. 3(a), but this time for the discrete spectrum of the deterministic sine wave seasonal cycle. In this situation the sign of the seasonal bias is unknown. It appears in the error spectrum as power at frequency zero.

If the timing of the production phase of the signal carriers is known precisely, $\Delta\phi_c = 0$, then the white noise variance is
equal to the variance of the piece of the sine wave that describes the portion of the year in which the carriers are produced. In this case the sign and value of the bias are completely "known" given the parameters of the model, or put another way, the proxy record can be attributed to the correct season.

PSEM can handle intermediate situations where, for example, we can parametrise with a proxy season length, $\tau_p$, an expected phase, $\langle\phi_c\rangle$, (the midpoint of the proxy production season), and an uncertainty in this phase, $0 < \Delta\phi_c < 2\pi$. In this situation
there is both a bias with an expected value and sign, and an uncertainty around this expected value that comes from the phase uncertainty.

Finally, if we include amplitude modulation of the seasonal cycle over the course of a precessionary cycle, $\sigma_a^2 > 0$, the size of any seasonal bias and bias uncertainty will vary over time. In the spectral domain this manifests as leakage of power from frequencies lower than $1 / T$ to higher frequencies and creates additional timescale-dependent errors corresponding to
uncertain changes in the magnitude of seasonal biases between time periods. Additionally, the magnitude of aliased seasonal cycle variation will vary over a precessionary period (see section 3.2 of part I).

## 2.5   Measurement error and individual variation

As, to a first order, the measurement error can be assumed to be independent between measurements, we simply add the power spectrum of a white noise error term $\sigma_{meas}$. More complex measurement errors such as machine drift or memory
between measurements could be integrated by adding a power spectrum characterising these machine characteristics. We add an additional error term, $\sigma_{ind}$, to account for inter-individual variation in the encoded signal. This is a catch-all term to include things like differences in depth habitat occupied by individuals, variation in the encoding of the signal, i.e. "vital effects" (Haarmann et al., 2011; Schiffelbein and Hills, 1984; Sadekov et al., 2008; Duplessy et al., 1970). For a given proxy measurement, the variance of this term is scaled by the number of individual signal carriers in the sample, $N$.

## 2.6   Calibration error

Finally we add uncertainty in proxy's calibration to temperature as a constant error, applying to all values in a given record, for which we don't know the sign but do have some idea of the magnitude $\sigma_{cal}$. This could for example be the standard error of the intercept term in a linear calibration model. This is implemented as additional power at frequency zero.

## 2.7   Power spectrum of the total error

As the individual error components are independent, their power spectra can be added together to get the spectrum of the total error. Once the spectral error model has been parametrised for a given core, proxy, and sampling scheme, the resulting empirical





power spectrum provides the magnitude and full temporal correlation structure of the error components. From this a number of useful quantities can be obtained. These include the error on individual proxy measurements, the error after smoothing a record to a lower resolution, the error on a time-slice and on the difference between two time-slices, the expected correlation between replicated proxy records. We illustrate these applications in the following two sections.

## 5  3   Illustration of the error spectrum approach for a hypothetical proxy record.

We first illustrate the error spectrum approach for a hypothetical 10 kyr foraminiferal Mg/Ca record. Parameter values have been chosen to be realistic while ensuring that all components of the error model are presented. We parametrise the climate spectrum as described in section 2.1, assuming a surface dwelling foraminifera at a virtual core position of 20° N, -18 E, calcifying between the surface down to 120 m (Fig. 2). We further assume that this taxon forms tests for a 7 month period of

the year, centered around the peak of the seasonal cycle but with an uncertainty in this phase of 2 months in either direction. A bioturbation depth of 10 cm and sedimentation rate of 10 cm kyr$^{-1}$ are assumed. Thirty foraminifera are picked from contiguous 1 cm thick sediment slices so that the resulting record has a sampling interval, $\Delta t$, of 100 years. We assume a measurement error of 0.25°C and inter-individual variation of 1°C. All parameters are given in Table 1.

The power spectra of the individual error components, together with their sum and the assumed power spectrum of the

climate, are shown for this example parameter set in Fig. 4. Power at $\nu = 0$ corresponds to errors that are constant for a given proxy time-series and thus do not shrink as additional measurements are averaged together. Here it is composed of those parts of the seasonal bias and bias uncertainty that are constant over time (i.e. not orbitally modulated), and the calibration uncertainty.

The power spectral densities of the measurement error and individual variation components are horizontal lines, indicating that their power is independent of frequency, i.e. that these error components have the property of white noise. This applies

also to the two components of aliased / redistributed seasonal cycle and stochastic climate variation.

In contrast, the component due to bioturbation and sediment slice thickness smoothing shows strong frequency dependence. The error due to smoothing is proportional to the variation in the climate signal, therefore, as variation in the climate is larger at lower frequencies, the error due to smoothing also increases towards lower frequencies. This is true up until the point at which the frequency examined exceeds the width of the smoothing filter, at which point the error due to smoothing declines towards

lower frequencies.

The individual values in record are most likely interpreted as a kind of mean of the time interval between observations and so we set the implicit reference timescale to be the same as the sampling resolution ($\Delta t = 100$). For this example, the timescale of the bioturbation smoothing, $\tau_b$, is 1000 years. The large difference between these timescales implies a large error due to bioturbation smoothing. At frequencies above about 2 x the bioturbation filter width, the power of the bioturbation error is

equal to the power of the reference climate.

The timescale-dependent portions of the seasonal bias and seasonal bias uncertainty are due to orbital modulation of the amplitude of the seasonal cycle. As this is an approximately 23 kyr cycle, these errors only become large at timescales approaching 23 kyr.



**Table 1.** Parameters required for the spectral error model with their values in example 1 here, plus possible sources.

| Parameter | Value | Description | Source |
|---|---|---|---|
| $\Delta t$ | 100 | The sampling frequency of the proxy record [years] | Approximated by the mean sampling frequency of an irregular timeseries |
| $\tau_r$ | 100 | Interpreted timescale of the proxy timeseries [years] | Equal to $\Delta t$ unless explicitly estimated |
| $T$ | 10100 | Total length of the proxy record [years] | Odd multiple of $\Delta t$ closest to the length of proxy record |
| $\tau_b$ | 1000 | Age heterogeneity of signal carriers due to bioturbation [years] | The bioturbation depth (estimated) divided by the sedimentation rate, or age-heterogeneity estimated from replicated radiocarbon dates |
| $\tau_s$ | 100 | Thickness of a sediment slice from which signal carrier are extracted [years] | Sediment slice thickness divided by the sedimentation rate |
| $\tau_p$ | 7/12 | Proportion of the year during which signal carriers are created | Sedimentation trap data or predictions from a planktonic foraminifera model such as PLAFOM 2.0, FORAMCLIM, or FAME |
| $\langle \phi_c \rangle$ | 0 | Expected phase of the signal carrier production period relative to the seasonal cycle [$-\pi, \pi$]. | Sediment-trap data or predictions from a planktonic foraminifera model such as PLAFOM 2.0, FORAMCLIM, or FAME |
| $\Delta \phi_c$ | $2\pi/3$ | Uncertainty in the phase of the signal carrier production [$0, 2\pi$]. | |
| $N$ | 30 | No. of signal carriers per proxy measurement | No. of signal carriers per proxy measurement |
| $\sigma_c^2$ | 2.2 | Variance of the seasonal cycle [proxy units$^2$] | Calculated from the modern climatological amplitude of the seasonal cycle estimated from instrumental data e.g. HadSST or reanalysis data |
| $\sigma_a^2$ | 0.014 | Variance of the orbital modulation of the seasonal cycle amplitude | Inferred from orbital variation in incoming solar radiation |
| $\phi_a$ | $\pi/2$ | Phase of the proxy record in relation to the orbital solar radiation cycle | Frequency of orbital cycle being modelled, e.g. procession 1/23 kyr |
| $\sigma_{meas}$ | 1/4 | Measurement error [proxy units] | Reproducibility of measurements on real world material |
| $\sigma_{ind}$ | 1 | Inter-individual variation [proxy units] | Individual foraminifera studies |
| $\sigma_{cal}$ | 1/4 | Calibration error [proxy units] | Standard error of the intercept term of a calibration regression model |





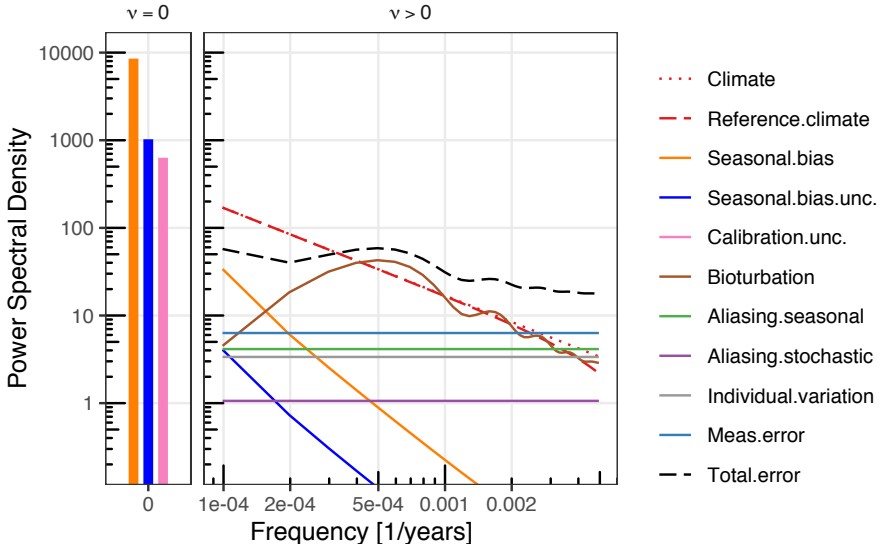

**Figure 4.** Power spectra of error components of a climate proxy record. As the proxy record was sampled at 100-year resolution only the power-law portion of the climate power spectrum is visible. The error spectra are plotted on log-log axes, with a broken frequency axis so that power at the zeroth frequency ($\nu = 0$) can be shown.

## 3.1 Timescale-dependent proxy error

The error for the individual proxy values at their original sampled resolution can be obtained by integrating the error spectrum. When the record is smoothed before its interpretation, the timescale represented by each point changes as does the error. The error for a given timescale can be obtained by integrating the error spectra after first multiplying with the transfer function of the smoothing filter (see section 4, Eq. 110 of part I).

Timescale-dependent error for the example parameter set is shown in Fig. 5 for timescales from 100 years (the original sampling resolution of the record) to 10000 years (a mean or time-slice of the entire length of the record), assuming a running mean smoothing filter. The errors are shown on the variance scale so that they are additive and can be plotted stacked together. The error(s) associated with the individual proxy measurements correspond to the rightmost edge of each subplot.

Fig. 5(a) includes error components that are constant for a given record and do not shrink as a record is smoothed. For example, a seasonal bias in a record due to signal carriers (e.g. Foraminifera) preferentially recording a particular part of the seasonal cycle will not disappear as additional proxy measurements are averaged together. For this example here, the total error is dominated by the constant part of the seasonal bias component, and to a lesser extent the bias uncertainty and calibration uncertainty.

Fig. 5(b) includes only those error components that are at least partly independent between timepoints and therefore vary with timescale, i.e. it excludes errors that originate from power at frequency zero. The components measurement error, individual variation, and the aliased components of the stochastic climate and seasonal cycle, all decline rapidly with timescale (i.e. as a





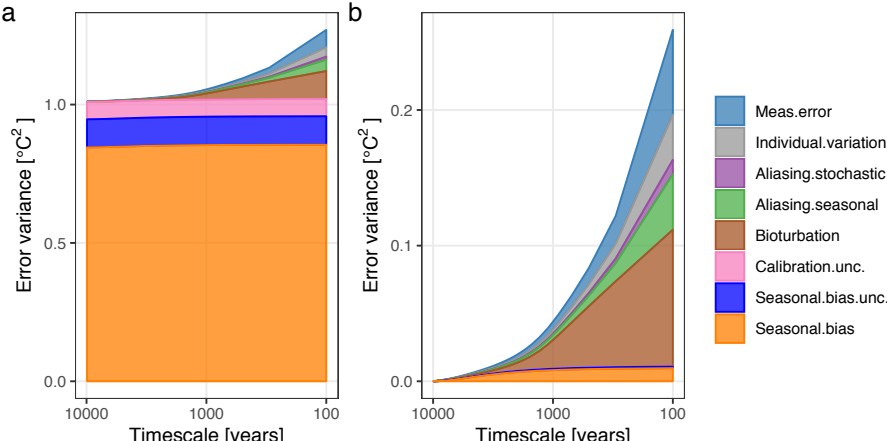

**Figure 5.** Timescale-dependent proxy error for an example parameter set. a) Timescale-dependent error variance for all components of spectral error model, b) Timescale-dependent error variance for all components, excluding those that manifest as constant errors that do not change with timescale, i.e. those originating as power at frequency zero.

record is smoothed with a running mean to lower resolutions), as the errors are independent between samples and so decline inversely with the number of samples being average together. The bioturbation component declines more slowly as errors are positively autocorrelated up until timescales of approximately $2\tau_b$. A portion of each of the seasonal bias and bias uncertainty components does vary with timescale due to orbital modulation of the amplitude of the seasonal cycle. They may become
important when comparing proxy values from two time-slices that are far enough apart in time that any seasonal bias may be differ between the two time-periods.

### 3.2 Error on a time-slice mean and the difference between two time-slices.

The information contained in the power spectra also allows us to estimate the uncertainty in the difference in the climate between two timepoints, or between two time-slices. A "time-slice" refers to an average taken over a set of proxy values
within a certain time period of interest. For example, using the parametrisation from Table 1, if we wanted to compare the mean climate over the first and last 1000 years the proxy record, the error variance on each individual time-slice would be the value at timescale = 1000 years in Fig. 5. If all the errors were independent in time then the error variance on the difference between these two time-slices would simply be the sum of the two variances. However, as some of these error components are autocorrelated (or even constant over the entire time-series) the covariance in the errors for the two time-slices needs to be
accounted for. The information to do this is contained in the power spectrum of the error (see section 4, Eq. 111-112 of part I). The uncertainty, or error, on the estimate of the difference between two time-slices is much smaller than the errors on the time-slices themselves (Fig. 6).




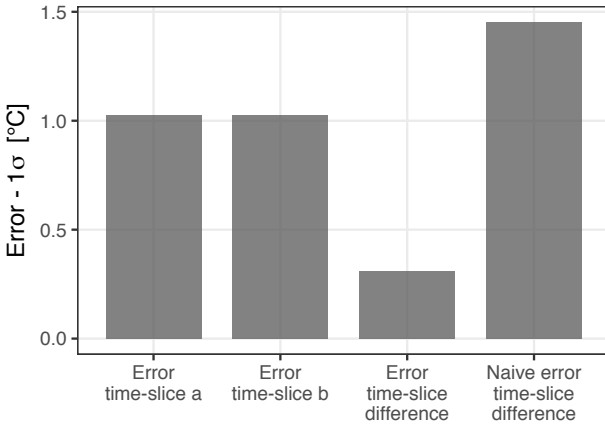

**Figure 6.** The uncertainty or error on the estimate of the mean climate over two time-slices covering the first and last 1000 years of the pseudo-proxy record, and the error in the estimate of the difference between these two time-slices. The realistic error estimate using PSEM (third column) is much smaller than the naive error estimate that one would obtain by just adding up the variances.

## 4 A worked example: replicated Holocene $\delta^{18}$O records from South Java

Here we illustrate the use of the error spectrum model on a real proxy record by applying it to replicated foraminiferal $\delta^{18}$O records taken from core GeoB 10054-4 off South Java in the Indian Ocean collected during the R/V SONNE 184 expedition (8°40.90'S, 112°40.10' E; 1076 m water depth; Hebbeln and et al, 2006). Replicated $\delta^{18}$O records were created using two dif-
5 ferent sampling schemes. In record 1 (Rep1) measurements were made on samples consisting of 5 *Globigerinoides ruber*(*s.s.*) tests each, at a mean interval of 83 years; in Rep2, 30 tests of *G. ruber*(*s.s.*) were used per sample at a mean interval of 246 years. An age model was constructed using 9 AMS-[14]C dates on mono-specific samples of *Trilobatus sacculifer* (S1). The Marine13 radiocarbon calibration curve was used to calibrate the ages and construct a linear age model (Reimer et al., 2013). As both records come from the same core, more advanced age-depth modelling is not required here.
A replicated set of radiocarbon dates taken on ten samples each of 10 foraminifera indicated an inter-individual standard deviation in age of 720 years (Dolman et al., in prep.) (S1), which we use for $\tau_b$, corresponding to a bioturbation depth of about 14 cm.

### 4.1 Parametrisation

To assist potential users of PSEM here we describe the parameter choices step by step.
Formally, the spectral error model describes only regularly sampled time-series whose total length $T$ is an odd multiple of of the sampling interval $\Delta t$. When applying the spectral error model to real proxy time-series we have to make some additional approximations to accommodate their inevitably irregular (in time) sampling intervals. Hence we approximate the sampling interval for each record $\Delta t$ by $\overline{\Delta t}$.





For the climate spectrum we again used a spliced empirical and theoretical spectrum and estimated the amplitude of the seasonal cycle as described in section 2.1, this time for the surface down to 50 m, resulting in an amplitude of 0.53 permille.

At this location *G. ruber* is thought to produce tests at an approximately equal rate throughout the year and represent the annual mean surface temperature in this region (Mohtadi et al., 2011). We therefore set $\tau_p = 1$, which implies year-round

production of signal carriers. As $\tau_p = 1$, the parameters $\langle \phi_c \rangle$ and $\Delta \phi_c$, which control the phase of signal carrier production relative to the seasonal cycle, have no effect on the error spectrum. Similarly, orbital modulation of the amplitude of the seasonal cycle will have only a small effect and is ignored here.

The sediment accumulation rate estimated from the calibrated radiocarbon dates and retrieval depths is approximately 20 cm/kyr. The sediment slices were 1 cm thick so $\tau_s$ was set to 50 years. A replicated set of radiocarbon dates taken on ten

samples each of 10 foraminifera indicated an inter-individual standard deviation in age of 720 years (unpublished data), which we use for $\tau_b$, corresponding to a bioturbation depth of about 14 cm.

For $\sigma_{meas}$ we use the analytic replicability of 0.1 permille. $\sigma_{ind}$ is more difficult to parametrise, we use 0.32 permille which was estimated by Sadekov et al. (2008) as the contribution of "vital effects" to replicability estimates for *G. ruber*.

### 4.2   Timescale-dependent uncertainty

Total error variance at the highest frequency resolved ($\overline{\Delta t}$) is much higher for Rep1 than Rep2 as there are fewer foraminifera per sample so that the individual variation, aliased seasonal cycle and aliased stochastic climate components are all larger for Rep1 than for Rep2 (Fig. 7). The effect of this can be seen clearly in Fig. 8(a) which shows Rep1 and Rep2 at their original irregularly sampled time-points together with their PSEM estimated uncertainties. Rep1 shows much higher variance despite the fact that Rep1 and Rep2 come from the same sediment core and therefore both experienced the same climate signal and the

same degree of bioturbation smoothing.

In Fig. 8(b), Rep1 and Rep2 have both been interpolated and smoothed to a regular 492 year resolution (492 = 2 x $\Delta t$ for Rep2). As the original time-series were irregular, a different number of proxy measurement now contribute to each mean value. To account for this, we evaluate PSEM separately for each point, setting ($\overline{\Delta t}$) to the timescale $\tau_{smooth}$ divided by the number of original proxy measurements. After this smoothing, the two series are in much closer agreement. The error for Rep1 has

shrunk much more than that for Rep2. In fact, smoothed to a timescale of 492 years, the error on Rep1 is now smaller than that for Rep2, due to the larger number of proxy measurements contributing on average to each point in the smoothed series (dotted vertical line in Fig. 7).

### 4.2.1   Expected correlation with the true climate

Finally, the property of a proxy record we are perhaps most interested in is its correlation with the true climate. The proxy and

climate can have low correlation due to a combination of non-climate variation (noise) in the proxy record and because variation in the climate has been smoothed out in the proxy record. As the noise and degree of smoothing are timescale dependent, so to is the proxy-climate correlation. Using the power spectra of the errors and the assumed power spectrum of the climate signal we can calculate the expected timescale dependent correlation between the proxy and climate. We do not in general know





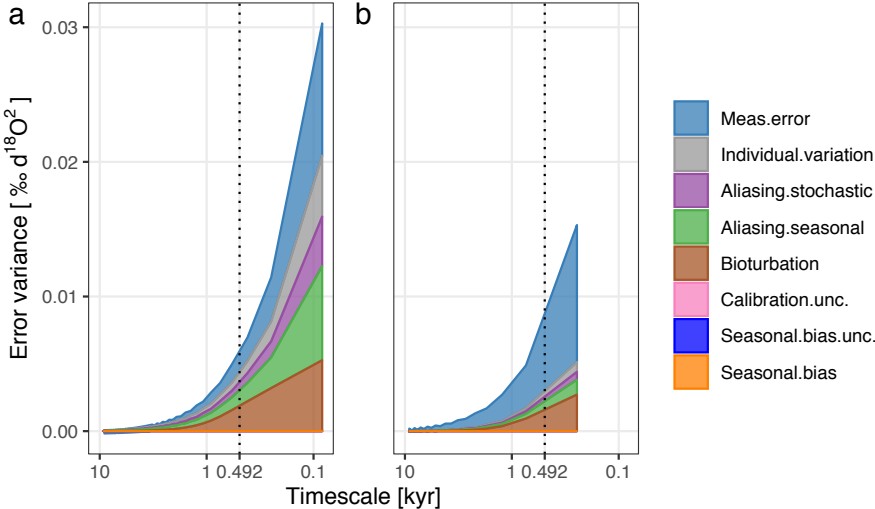

**Figure 7.** Timescale-dependent error variance for two different $\delta^{18}$O sampling strategies at GeoB 10054-4. A) Five foraminifera per measured sample with a mean time interval of 83 years between samples, B) Thirty foraminifera per measured sample with a mean time interval of 246 years between samples. The vertical line at 0.492 kyr indicates the timescale to which the proxy records are smoothed in Fig. 8b.

the true climate and so cannot test the accuracy of the climate-proxy correlation estimate, however, we can also calculate the expected correlation between replicate proxy records and use this as a partial test of the model, under the assumption that only the processes considered in PSEM affect the proxy record. The results (Fig. 9) indicate an increasing expected correlation from around 0 at centennial timescales to around 0.5 at millennial timescales. This is an upper bound estimate as the chronological
uncertainty and other effects not considered here will further decrease the climate to proxy relationship.

We estimated the timescale dependent correlation between Rep1 and Rep2 using the R package corit (Reschke et al., 2019). The irregular time-series were first interpolated to high resolution regular timeseries and then smoothed with a set of increasingly wide running mean filters before calculating the correlation between them. The observed correlation between Rep1 and Rep2 is somewhat higher than that estimated from the error spectra (Fig. 9), perhaps indicating that we have assumed for
example slightly too high measurement error, although it is unclear if this difference is statistically significant. At timescales above 1000 years estimates of the observed correlation become very variable due to there being very few effective data points left after smoothing.

## 5  Discussion and conclusions

Understanding the errors associated with climate proxies is an important task for paleoclimate research. Proxy errors, defined
here as differences between the inferred climate and the unknown true climate, can be large and can thus strongly influence our understanding of past climate history and the functioning of the climate system. Many components of proxy error have

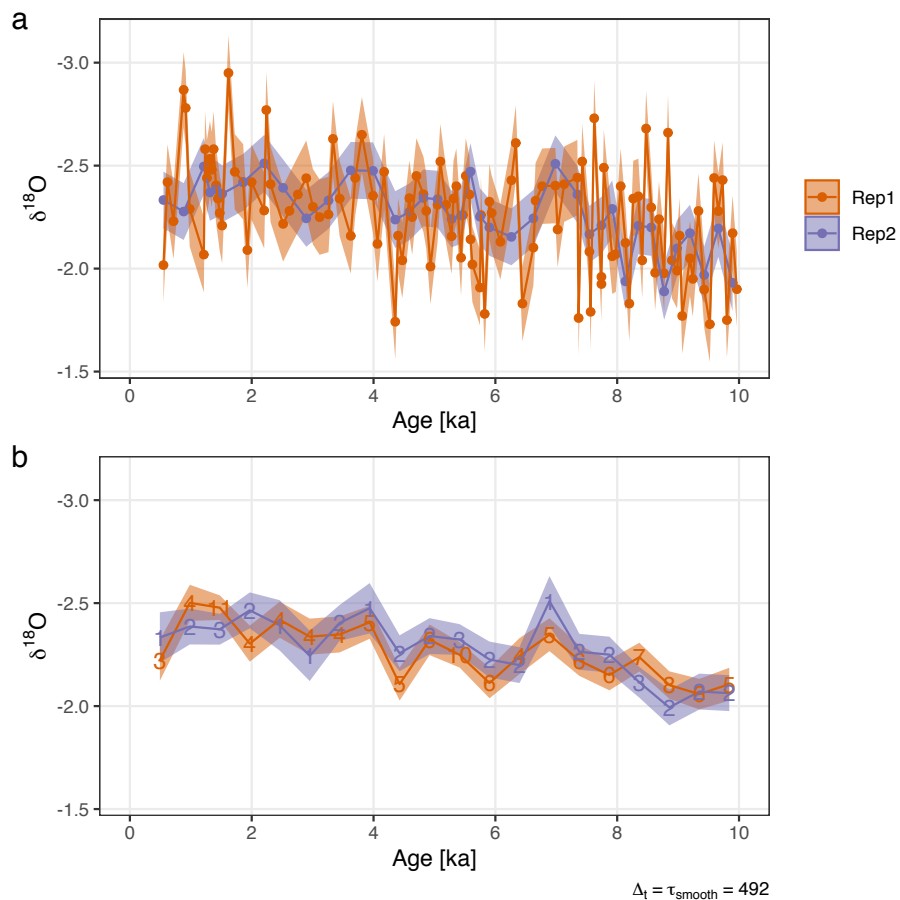

**Figure 8.** $\delta^{18}O$ records from GeoB 10054-4, (a) at their original temporal resolutions, (b) interpolated to a regular time-series and smoothed to 492 year resolution. Shaded regions show 1-sigma error estimated using the PSEM. The error due to smoothing by bioturbation is excluded as it is deterministic and thus the same for each record. Numbers indicate the number of original data points contained in each averaged proxy value.

a complex temporal autocorrelation structure, making them timescale dependent and a challenge to properly quantify and account for.

The model introduced here (PSEM) and in the companion paper (Kunz et al., 2020) offers a rigorous and compact way to calculate and express this structure as error spectra, specifically here in this first version for marine sediment cores. Once

5  defined, the error spectra can be used to calculate many quantities that will be useful to paleoclimate research. In addition to the error on individual measurements, these include the error after smoothing the record, the error on time-slices and differences between time-slices, the expected correlation between replicates of a proxy record and between a record and the true climate.

As with every model, some challenges remain, in particular how to deal rigorously with the irregularity of real proxy time-series, the climate dependency of the habitat of the recording organisms and the error associated with the age-uncertainty.





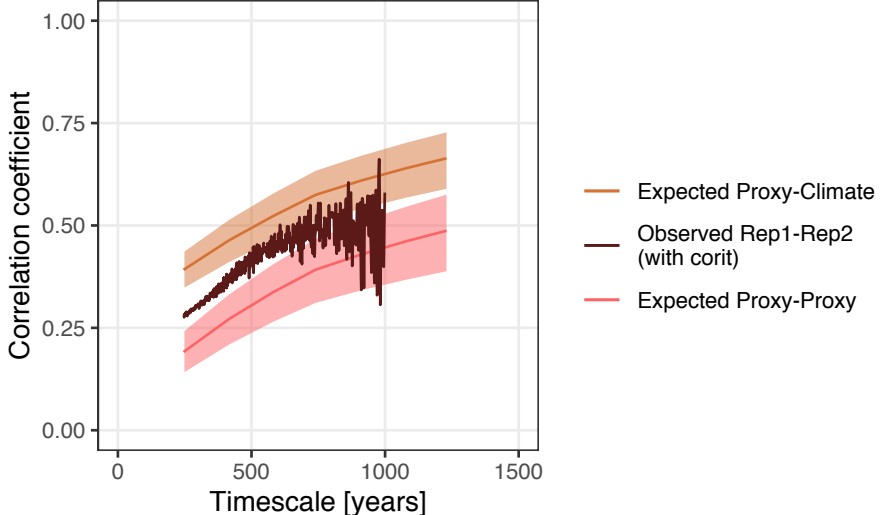

**Figure 9.** Timescale-dependent correlation between replicated $\delta^{18}$O records at GeoB 10054-4 and their expected correlation with each other and with the true climate. The dark brown curve shows the observed timescale-dependent correlation between the replicate proxy records Rep1 and Rep2. The light brown and pink curves show the expected correlation between the two proxy records and between the proxy records and the true climate as calculated from the error spectra. The shaded areas indicate +- the expected standard deviation in the correlation coefficients.

Nevertheless, we argue that the spectral error approach represents a significant advance towards obtaining reliable uncertainty estimates for all quantities of interest rather than single estimates applied uniformly to the record.

Simulation based forward modelling approaches such as sedproxy (Dolman and Laepple, 2018) and PRYSM (Dee et al., 2015) could also be used to estimate these quantities by generating and summarizing over many simulated pseudo-proxy records. The advantages of PSEM are that it provides an analytic understanding of the timescale dependence of error components while retaining the mechanistic understanding of the proxy generating process and can make these uncertainty estimates rapidly for large sets of parameters. For example, to directly model the error for a wide range of potential sediment core characteristics (sedimentation rate and bioturbation depth), with different sampling schemes and at different locations with differing climate properties such as seasonal cycle amplitude. This allows us to both better interpret existing proxy record and to optimize future field work to answer specific questions.

Beyond modelling errors, PSEM also facilitates the use of the proxy variability itself to make inferences about the climate system. It allows prediction of the variability observed in individual foraminifera assemblages (IFA) (e.g., Groeneveld et al., 2019; Thirumalai et al., 2019) and thus to directly test the sensitivity of IFA statistics on the sedimentation rate, seasonality or the spectrum of climate variability. Finally, PSEM provides the basis to develop spectral correction approaches that infer the climate spectrum from the corrupted and distorted proxy spectrum, building on the approaches previously proposed for simpler sediment models (Laepple and Huybers, 2013) or for aliasing only (Kirchner, 2005).





The PSEM version proposed here includes the sediment proxy processes described earlier for the proxy forward model sedproxy (Dolman and Laepple, 2018) and represents a trade-off between complexity and completeness. For example, the interaction of seasonality in the recording and climate signal is the only slowly varying process so far included. However, the general formulation of the PSEM allows to other processes to be added. For example, depending on the timescale of interest and the proxy type, other slowly varying processes such as long-term changes in seawater Mg/Ca (Medina-Elizalde et al., 2008), or long-term instrumental drift and memory effects of the measurement process, could be included by specifying the power spectra. When accounting for these processes, the use of PSEM vs. the classical single value uncertainty approach becomes even more important.

Here, and in part I, we have defined analytical expressions specifically for sediment archived climate proxies; however, the approach is applicable to other proxy types as most proxy types experience similar error generating and distorting processes. For example, smoothing also affects water isotopes measured in ice-cores via water vapor diffusion (Johnsen et al., 2000) and geochemical indices measured in coral records (Gagan et al., 2012) via successive incremental calcification in corals. These processes can also be expressed as filters on the climate signal and the power spectra of the errors they produce derived in a similar way. Thus, we hope that PSEM presents an important step towards providing more realistic error estimates for paleoclimate research.

*Code and data availability.* The Proxy Spectral Error Model (PSEM) is implemented as an R package. Its source code is available from the public git repository https://github.com/EarthSystemDiagnostics/psem. Down-core radiocarbon dates for GeoB 10054-4 are provided in Supplement S1. $\delta^{18}O$ isotope data will be archived in PANGAEA on publication.

*Author contributions.* TK, AMD and TL designed the conceptual outline of the research. AMD coded the PSEM R package and produced
5   the figures. AMD and TL wrote the manuscript with contributions from TK and JG.

*Competing interests.* The authors declare that they have no conflict of interest.

*Acknowledgements.* This work was supported by German Federal Ministry of Education and Research (BMBF) as Research for Sustainability initiative (FONA); www.fona.de through PALMOD project (FKZ: 01LP1509C). TL and TK have received funding from the European Research Council (ERC) under the European Union's Horizon 2020 research and innovation programme (grant agreement no. 716092). The
10   work profited from discussions at the CVAS working group of the Past Global Changes (PAGES) programme. We thank Henning Kuhnert for isotope analyses, Mahyar Mohtadi for providing the sediment core GeoB 10054-4, Nele Behrendt and Lena Kafemann for processing samples, and Gesine Mollenhauer and Torben Gentz for radiocarbon dating.



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
