# Peer review of "A spectral approach to estimating the timescale-dependent uncertainty of paleoclimate records — Part II: Application and interpretation."

_Climate of the Past, 2019_

## Referee Comment (RC1) · Anonymous Referee #1 · 17 May 2020

Earth's climate exhibits variability on a wide range of different time scales, and temperature time series have significant serial correlations. A 1/f noise can roughly approximate temperature fluctuations in the Holocene up to the industrial age. Applying pseudo-proxy methods to correlated signals, the authors demonstrate that proxy errors depend on the time scale, and they develop a framework for estimating the scale-dependent (frequency-dependent) proxy errors. The method is well-motivated, and it seems very practical, with an associated R-package. In my view, the paper represents an essential contribution to the field, and I congratulate the authors on having written an excellent article.

[Figure]

I have one general suggestion and a few minor comments.

My general suggestion is that the authors explore how different assumptions on the characteristics of the climate fluctuations affect their results. I do not think it is necessary to do this systematically, but it should be straight forward to plug in a few examples. For instance, different power laws and maybe the spectrum with two characteristic time scales (the spectrum of a two-box model driven by white noise). It would also be interesting to see a discussion of the applicability of the method beyond the Holocene climate.

My minor comments are the following:

The authors state that "Currently the temporal covariance structure of proxy uncertainties is largely ignored in the literature". This is true, but there are a few papers. For instance this one: Nilsen et al., Assessing the performance of the BARCAST climate field reconstruction technique for a climate with long-range memory, Climate of the Past, 2018. On line 24: "The power-spectrum of a proxy error contains all the information required to derive timescale dependent uncertainties." My comment is that, yes this is true, if you only consider second-order statistics. In principle, there can be other sources of uncertainty; for instance, changes in the fluctuation level over time. Line 18: calcite should not be in italics, same on line 19, 21, 28, and in Table 1.

---

## Referee Comment (RC2) · Anonymous Referee #2 · 31 Aug 2020

The manuscript demonstrates how the theoretical framework presented in the first companion paper can be used to estimate proxy uncertainties that are timescale-dependent and correlated in time using the example of marine sediment cores. The paper comes with an R-package that makes the methodology easily accessible. The associated package was easy to install, and the functions within the package are well documented and easy to use.

In my opinion, the manuscript is well-written, scientifically sound and provides a great contribution to the field. I agree with the comments of Referee #1 in that it would be interesting to see how different assumptions on the characteristics of the climate

fluctuations affect their results, and a discussion of the applicability of this approach beyond the Holocene.

I have the following minor comments and suggestions:

Text in subscript should not be in italics: page 4 line 17, page 5 line 1, page 8 lines 19, 21 and 27, page 14 lines 12 and 23, and in Table 1. Page 8 line 9: parenthesis should generally not be preceded by a comma. Page 4 line 2: "an heuristic" should be changed to "a heuristic". Page 13 line 14: 'of' is written twice in "odd multiple of of the sampling interval". Page 18 line 4: delete 'to' in "PSEM allows to other processes".

I also have a few technical comments on the package.

Some arguments are not included in the argument description of their respective functions: OrbitalError.Rmd: delta_phi_c OrderStages: rev ProxyErrorSpectrum: n.k, delta_phi_c, sigma.cal S_E: n.k sinc: normalized

The argument 'tau_a' is not included in /usage in the documentation file of 'ProxyError-Spectrum'.

Generally, data sets located in the /data folder in an R-package should be documented. However, since these files seem to not be intended to be used directly by the end-user, but rather used or processed by other scripts, I would suggest, if possible, to place these files in another folder.

---

## Author Comment (AC1) · 25 Sep 2020

*Extracts of referee's comments appear in italics*

Dear Referee,

Thank you for taking the time to review our work and for your positive comments and suggestions.

*My general suggestion is that the authors explore how different assumptions on the characteristics of the climate fluctuations affect their results. I do not think it is necessary to do this systematically, but it should be straight forward to plug in a few examples. For instance, different power laws and maybe the spectrum with two characteristic time scales (the spectrum of a two-box model driven by white noise).*

We agree that the manuscript would benefit from some exploration of how changes to the assumed power-spectrum of climate affect the estimated time-scale dependent errors.

In brief: if the power of the high-frequency portion of the spectrum (about which we know something from the instrumental record) is held constant, while the slope at lower frequencies is made steeper, this increases the error components due to bioturbation - the smoothing effect and also the amount of climate variation redistributed as white noise. There is however some interaction with the parameter $tau_b$, which controls the amount or depth of sediment mixing and therefore the timescales integrated by the bioturbation filter. The deeper the mixing, the larger the effect of varying the power-spectrum slope. If the power at high frequencies is not keep constant, e.g. if using just a pure power-law spectrum, where changing the slope also effects power at high frequencies, then this interaction with $tau_b$ gets more complicated, as a shallower slope can mean more power at high frequencies.

We will add a section exploring these effects either as part of the main manuscript or as a supplemental section.

*It would also be interesting to see a discussion of the applicability of the method beyond the Holocene climate.*

Regarding the application of this method beyond the Holocene. Many of the error components, such as the bioturbation smoothing and seasonal aliasing, should remain approximately correct; however if we include glacial-interglacial cycles there will be

larger variations in both the sedimentation rate and the seasonality of the signal carriers (e.g. foraminifera). For the seasonal cycle of the climate, the amplitude of the seasonal cycle and the precession driven modulation of the seasonal cycle will vary with the longer inclination and eccentricity orbital cycles – although the proportional changes are relatively small.

For the assumed stochastic climate spectrum, the key issue is the assumption of stationarity. If multiple glacial cycles are included then one could argue that the spectrum is again stationary and still dominated by a power-law type variation. It becomes more difficult to justify if just one glacial-interglacial is included. In summary, we would argue that current approach also works beyond the Holocene, albeit less accurately than within the Holocene. Nonetheless it is a significant improvement over assuming independent errors. We propose to discuss these issues in the manuscript.

Minor comments:

*The authors state that "Currently the temporal covariance structure of proxy uncertainties is largely ignored in the literature". This is true, but there are a few papers. For instance this one: Nilsen et al., Assessing the performance of the BARCAST climate field reconstruction technique for a climate with long-range memory, Climate of the Past, 2018.*

We will add references to the existing literature paper that does consider the temporal covariance in proxy errors.

*On line 24: "The power-spectrum of a proxy error contains all the information required to derive timescale dependent uncertainties." My comment is that, yes this is true, if you only consider second-order statistics. In principle, there can be other sources of uncertainty; for instance, changes in the fluctuation level over time.*

This statement should be qualified by the necessary assumption of stationarity, which we make clear in our companion part I article, but we should also make clear here in part II.

*Line 18: calcite should not be in italics, same on line 19, 21, 28, and in Table 1.*

We will fix this error.

---

## Author Comment (AC2) · 25 Sep 2020

*Extracts of referee's comments appear in italics*

Dear Referee,

Thank you for taking the time to review our work and for your positive comments and suggestions.

[Figure]

*I agree with the comments of Referee 1 in that it would be interesting to see how different assumptions on the characteristics of the climate fluctuations affect their results, and a discussion of the applicability of this approach beyond the Holocene.*

Regarding the assumptions on the characteristics of the climate fluctuations and discussion of applicability outside of the Holocene, we answer here as for reviewer 1.

We agree that the manuscript would benefit from some exploration of how changes to the assumed power-spectrum of climate affect the estimated time-scale dependent errors.

In brief: if the power of the high-frequency portion of the spectrum (about which we know something from the instrumental record) is held constant, while the slope at lower frequencies is made steeper, this increases the error components due to bioturbation - the smoothing effect and also the amount of climate variation redistributed as white noise. There is however some interaction with the parameter $tau_b$, which controls the amount or depth of sediment mixing and therefore the timescales integrated by the bioturbation filter. The deeper the mixing, the larger the effect of varying the power-spectrum slope. If the power at high frequencies is not keep constant, e.g. if using just a pure power-law spectrum, where changing the slope also effects power at high frequencies, then this interaction with $tau_b$ gets more complicated, as a shallower slope can mean more power at high frequencies.

We will add a section exploring these effects either as part of the main manuscript or as a supplemental section.

Regarding the application of this method beyond the Holocene. Many of the error components, such as the bioturbation smoothing and seasonal aliasing, should remain approximately correct; however if we include glacial-interglacial cycles there will be larger variations in both the sedimentation rate and the seasonality of the signal carriers (e.g. foraminifera). For the seasonal cycle of the climate, the amplitude of the seasonal

cycle and the precession driven modulation of the seasonal cycle will vary with the longer inclination and eccentricity orbital cycles – although the proportional changes are relatively small.

For the assumed stochastic climate spectrum, the key issue is the assumption of stationarity. If multiple glacial cycles are included then one could argue that the spectrum is again stationary and still dominated by a power-law type variation. It becomes more difficult to justify if just one glacial-interglacial is included. In summary, we would argue that current approach also works beyond the Holocene, albeit less accurately than within the Holocene. Nonetheless it is a significant improvement over assuming independent errors. We propose to discuss these issues in the manuscript.

Minor comments:

*Text in subscript should not be in italics: page 4 line 17, page 5 line 1, page 8 lines 19, 21 and 27, page 14 lines 12 and 23, and in Table 1.*

*Page 8 line 9: parenthesis should generally not be preceded by a comma.*

*Page 4 line 2: "an heuristic" should be changed to "a heuristic".*

*Page 13 line 14: 'of' is written twice in "odd multiple of of the sampling interval".*

*Page 18 line 4: delete 'to' in "PSEM allows to other processes".*

We will correct these textual errors, thank you.

*I also have a few technical comments on the package.*

*Some arguments are not included in the argument description of their respective functions:*

*OrbitalError: "delta_phi_c"*

*OrderStages: rev*

*ProxyErrorSpectrum: n.k, "delta_phi_c", sigma.cal, "S_E": n.k*

*sinc: normalized*

*The argument "tau_a" is not included in /usage in the documentation file of 'ProxyError-Spectrum'.*

These arguments should now be defined in all the relevant help files. tau_ has been removed from the function definition as the period of the orbital cycle is parametrized by its inverse, nu_a, the corresponding frequency.

*Generally, data sets located in the /data folder in an R-package should be documented. However, since these files seem to not be intended to be used directly by the end-user, but rather used or processed by other scripts, I would suggest, if possible, to place these files in another folder.*

We will move these data to /R/sysdata.rda which is more appropriate for internal data.

---

## Author Response (AR1)

**Response to reviewers' comments:**

**Extracts of comments are in italics.**

**Reviewers 1 and 2 had the same comments about the climate spectrum assumptions and the applicability of the method beyond the Holocene.**

*My general suggestion is that the authors explore how different assumptions on the characteristics of the climate fluctuations affect their results. I do not think it is necessary to do this systematically, but it should be straight forward to plug in a few examples. For instance, different power laws and maybe the spectrum with two characteristic time scales (the spectrum of a two-box model driven by white noise).*

We agree that the manuscript would benefit from some exploration of how changes to the assumed power-spectrum of climate affect the estimated time-scale dependent errors.

In brief: if the power of the high-frequency portion of the spectrum (about which we know something from the instrumental record) is held constant, while the slope at lower frequencies is made steeper, this increases the error components due to bioturbation - the smoothing effect and also the amount of climate variation redistributed as white noise. There is however some interaction with the parameter tau_b, which controls the amount or depth of sediment mixing and therefore the timescales integrated by the bioturbation filter. The deeper the mixing, the larger the effect of varying the power-spectrum slope. If the power at high frequencies is not keep constant, e.g. if using just a pure power-law spectrum, where changing the slope also effects power at high frequencies, then this interaction with tau_b gets more complicated, as a shallower slope can mean more power at high frequencies.

We have added an appendix to the manuscript exploring these effects. This can be moved to a supplement if the editor find this more appropriate.

*It would also be interesting to see a discussion of the applicability of the method beyond the Holocene climate.*

Regarding the application of this method beyond the Holocene. Many of the error components, such as the bioturbation smoothing and seasonal aliasing, should remain approximately correct; however if we include glacial-interglacial cycles there will be larger variations in both the sedimentation rate and the seasonality of the signal carriers (e.g. foraminifera). For the seasonal cycle of the climate, the amplitude of the seasonal cycle and the precession driven modulation of the seasonal cycle will vary with the longer inclination and eccentricity orbital cycles – although the proportional changes are relatively small.

For the assumed stochastic climate spectrum, the key issue is the assumption of stationarity. If multiple glacial cycles are included then one could argue that the spectrum is again stationary and still dominated by a power-law type variation. It becomes more difficult to justify if just one glacial-interglacial is included.

In summary, we would argue that current approach also works beyond the Holocene, albeit less accurately than within the Holocene. Nonetheless it is a significant improvement over assuming independent errors.

We have added a paragraph to the discussion starting on page 17 line 12.

**Reviewer 1. Minor comments:**

*The authors state that "Currently the temporal covariance structure of proxy uncertainties is largely ignored in the literature". This is true, but there are a few papers. For instance this one: Nilsen et al., Assessing the performance of the BARCAST climate field reconstruction technique for a climate with long-range memory, Climate of the Past, 2018.*

We have added reference to Moberg and Brattström, 2011 (p2. Line 11) who deal with autocorrelated errors. However, while Nilsen et al. deal with long-range memory in the climate field, the errors they model are white and therefore not timescale dependent.

*On line 24: "The power-spectrum of a proxy error contains all the information required to derive timescale dependent uncertainties." My comment is that, yes this is true, if you only consider second-order statistics. In principle, there can be other sources of uncertainty; for instance, changes in the fluctuation level over time.*

This statement should be qualified by the necessary assumption of stationarity, which we make clear in our companion part I article, but we should also make clear here in part II. We have altered line 25 on page 2 to make this condition clear.

*Line 18: calcite should not be in italics, same on line 19, 21, 28, and in Table 1.*

We have fixed this and similar formatting errors throughout the manuscript.

**Reviewer 2. Minor comments**

*Text in subscript should not be in italics: page 4 line 17, page 5 line 1, page 8 lines 19, 21 and 27, page 14 lines 12 and 23, and in Table 1.*

These formatting errors have been corrected throughout the manuscript.

*Page 8 line 9: parenthesis should generally not be preceded by a comma.*

Corrected.

*Page 4 line 2: "an heuristic" should be changed to "a heuristic".*

Corrected.

*Page 13 line 14: 'of' is written twice in "odd multiple of of the sampling interval". Page 18 line 4: delete 'to' in "PSEM allows to other processes".*

Corrected.

*I also have a few technical comments on the package.*

*Some arguments are not included in the argument description of their respective functions:*
*OrbitalError.Rmd: delta_phi_c*
*OrderStages: rev*
*ProxyErrorSpectrum: n.k, delta_phi_c, sigma.cal*
*S_E: n.k*
*sinc: normalized*

*The argument 'tau_a' is not included in /usage in the documentation file of 'ProxyError-Spectrum'.*

These arguments are now be defined in all the relevant help files. tau_a has been removed from the function definition as the period of the orbital cycle is parametrized by its inverse, nu_a, the corresponding frequency.

*Generally, data sets located in the /data folder in an R-package should be documented. However, since these files seem to not be intended to be used directly by the end-user, but rather used or processed by other scripts, I would suggest, if possible, to place these files in another folder.*

These data have been moved to /R/sysdata.rda which is more appropriate for internal data.

**Additional changes:**

Title:

The title of the companion paper had to be changed. We have re-written the title of this part II paper to be consistent with the title of part I.

New title: A spectral approach to estimating the timescale-dependent uncertainty of paleoclimate records – Part II: Application and interpretation.

Code and data availability:

A snapshot of the R package psem at the time we created the figures for this manuscript has been archived at Zenodo and the DOI added to Code and data availability section.

We added the down-core isotope data to Supplement S1. These data are part of a larger set that will be published in 2021 and deposited in Pangaea. There is currently a 6-month wait for Pangaea submissions. As we do not actually make any inferences about the past climate in this paper, and use the data only to illustrate our method, to avoid further delay we hope that providing the data here will be sufficient.

[revised manuscript text omitted]